# Parent Acceptance toward Inactivated COVID-19 Vaccination in Children with Acute Lymphoblastic Leukemia: The Power of Oncologist and Alliance

**DOI:** 10.3390/vaccines10122016

**Published:** 2022-11-25

**Authors:** Yifei Ma, Nianqi Liu, Guanqing Zhong, Dao Wang, Lu Cao, Shenrui Bai, Pengfei Zhu, Ao Zhang, Xinjia Wang

**Affiliations:** 1Department of Orthopedics and Spine Surgery, The Second Affiliated Hospital of Shantou University Medical College, 69 Dongsha North Road, Shantou 515000, China; 2Department of Bone and Soft Tissue Oncology, Cancer Hospital of Shantou University Medical College, 7 Raoping Road, Shantou 515041, China; 3Faculty of Psychology, Institute of Educational Science, Huazhong University of Science and Technology, Wuhan 430074, China; 4Department of Clinical Laboratory, State Key Laboratory of Oncology in South China, Collaborative Innovation Center for Cancer Medicine, Guangdong Key Laboratory of Nasopharyngeal Carcinoma Diagnosis and Therapy, Sun Yat-sen University Cancer Center, Guangzhou 510060, China; 5Department of Pediatrics, The First Affiliated Hospital of Zhengzhou University, Zhengzhou 450000, China; 6Department of Hematological Oncology, State Key Laboratory of Oncology in South China, Collaborative Innovation Center for Cancer Medicine, Sun Yat-sen University Cancer Center, Guangzhou 510060, China; 7Department of Clinical Laboratory, The First Affiliated Hospital of Zhengzhou University, Zhengzhou 450000, China

**Keywords:** COIVD-19 vaccine, parent acceptance, vaccination hesitancy, oncologist recommendation, patient–oncologist alliance, acute lymphoblastic leukemia

## Abstract

Objectives: The current study aims to survey the willingness of parents to vaccinate their children, who are childhood acute lymphoblastic leukemia survivors (CALLS), and identify factors associated with vaccine acceptance. Methods: Parents of CALLS on/off treatment, with the general condition of being amendable to vaccination, were recruited for interviews with attending oncologists about COVID-19 vaccination acceptance from July to November 2021 in China. After controlling for socioeconomic factors, the Association of Oncologists’ recommendations and parent–oncologist alliance with acceptance status were investigated. For validation, propensity score-matched (PSM) analysis was used. Results: A total of 424 families were included in the study, with CALLS mean remission age of 5.99 ± 3.40 years. Among them, 91 (21.4%) agreed, 168 (39.6%) hesitated, and 165 (38.9%) parents disagreed with the vaccination. The most common reason that kept parents from vaccinating their children was lack of recommendations from professional personnel (84/165, 50.9%), and massive amounts of internet information (78/175, 44.6%) was the main nonhealthcare resource against vaccination. Logistic regression analysis showed that only the recommendation from the oncologist was associated with parents’ vaccine acceptance (OR = 3.17, 95% CI = 1.93–5.20), as demonstrated by PSM comparison (42 in recommendation group vs. 18 in nonrecommendation group among 114 pairs, *p* < 0.001). An exploratory analysis revealed that parents with a better patient–oncologist alliance had a significantly higher level of acceptance (65.6% in alliance group vs. 15.6% in nonalliance group among 32 pairs, *p* < 0.001). Conclusions: Due to a lack of professional recommendation resources and the potential for serious consequences, parents were generally reluctant to vaccinate their CALLS. The recommendation of oncologists, which was influenced by the parent–oncologist alliance, significantly increased acceptance. This study emphasizes the critical role of oncologists in vaccinating cancer survivors and can be used to promote COVID-19 vaccines among vulnerable populations.

## 1. Introduction

Contraction of severe acute respiratory syndrome coronavirus 2 (SARS-CoV-2) has been shown to be disastrous for cancer patients, with pediatric patients with hematologic malignancies being the most vulnerable to such attacks [1,2]. Vaccination for healthy children against the virus has been steadily implemented worldwide. In November 2021, the Food and Drug Administration (FDA) approved the mRNA vaccination in children, and in China, the CoronaVac vaccination for healthy children was approved even earlier [3,4]. However, children with a history of hematological malignancies, such as acute lymphoblastic leukemia (ALL), are still not candidates [4,5]. Although previous research has shown that children who have had allogeneic hematopoietic stem cell transplants can mount protective immune responses to inactivated influenza vaccination [6,7], there has been little research on coronavirus disease 2019 (COVID-19) vaccination in children with hematological malignancies. Given the high rate of severe, fatal courses of COVID-19 infection in childhood ALL survivors (CALLS), it is critical to provide CALLS with vaccination when appropriate and available.

The literature revealed relatively high acceptance of vaccination for parents of healthy children, whereas research on hesitancy for parents of childhood cancer survivors was relatively lacking. Although data are still scarce, one previous study of 69 parents of children with hematological cancer revealed widespread hesitancy due to concerns about the safety and efficacy of COVID-19 vaccination [8]. Given the complex interaction with medical facilities even during cancer remission as well as the high rate of vaccination refusal in cancer patients overall, the data on parents’ attitudes toward COVID-19 vaccination may aid in understanding how to increase vaccination coverage [5].

According to safety data of inactivated, non-COVID-19 vaccines such as influenza vaccines, the general recommendation was that pediatric cancer survivors, regardless of chemotherapy status, could receive vaccination [9]. To be as cautious as possible, oncologists may advise vaccination one year after treatment in the complete remission phase free of medical surveillance [9]. Parental education by healthcare professionals, such as physicians, has been shown to significantly reduce vaccine apprehension [9]. A large-scale survey of 744 adult breast cancer survivors in China found that 74% of respondents were hesitant or refused to vaccinate, with the most common reason being a lack of a professional source of information about vaccination safety or efficacy of COVID-19 for cancer survivors. Notably, the person most likely to influence patient decisions was the doctor in charge of treatment [10].

Although childhood cancer patients’ oncologists have always been the source of vaccination information, providing information on COVID-19 vaccination would currently be difficult. On the one hand, providing informed recommendations may not always change the attitude. It has been demonstrated that the therapeutic alliance with the oncologists, which stands for the degree of mutual trust and understanding on both sides of health care, has an impact on parents’ adherence to child therapy [11]. It is critical to understand to what extent the physicians’ recommendations may influence the attitude of parents of childhood cancer patients, and how the patient–oncologist alliance plays its role requires investigation to tailor the approach of communication. On the other hand, previous evidence on the safety and efficacy of non-COVID-19 vaccines does not dispel concerns about COVID-19 vaccines due to the lack of large-scale clinical validation.

The current study aims to identify independent variables influencing approval or refusal of COVID-19 vaccination in parents of CALLS and then investigate how the patient–oncologist alliance mediates the relationship between recommendation and parental attitude.

## 2. Materials and Methods

### 2.1. Participant Enrollment

From 5 July 2021 to 29 November 2021, consecutive parents of CALLS were enrolled in the cross-sectional observational study in the outpatient clinical setting of a pediatric hematology department. Key inclusion criteria included: (a) are parents of surviving ALL patients, aged from 3 to 17 years old, who have been treated or are being treated in the First Affiliated Hospital of Zhengzhou University, (b) have oncologist-evaluated general conditions tolerable to traditional vaccines, and (c) do not have COVID-19 infection history. Key exclusion criteria included history of anaphylaxis to any component of approved traditional vaccines, inadequate literacy, and refusal to follow-up. Prior to enrollment, participants had to provide either verbal or written informed consent to participate in the study. The study was approved by the institutional review board of the Second Affiliated Hospital of Shantou University Medical College.

### 2.2. Patient Follow-Up and Instrument Evaluation

During the outpatient visit, parents of the CALLS underwent semistructured interviews to determine whether their family would be willing to receive the COVID-19 vaccine for the CALLS one year after remission, the attending oncologist’s recommendation status regarding the vaccination, and any outside influences that might affect their attitude toward the vaccination. If parents expressed concerns about the doctor’s stance on vaccination, they were advised to consult with the attending oncologist and a new outpatient interview was scheduled. The attending oncologists of the participants were blind to the research protocol, as we retrospectively interviewed the parents without seeking confirmation from their doctor. The standard question about the willingness of COVID-19 vaccination is “does your family agree, hesitate, or disagree to vaccinate your CALLS with COVID-19 vaccines one year after cancer remission?”, so attitudes were divided into “agree”, “hesitancy”, and “disagree”, and reasons for disagreement were collected then. The recommendation status toward COVID-19 vaccination by the attending oncologist of the CALLS was divided into “recommended” and “not recommended”, and neutral reactions were classified as “not recommended”. Furthermore, we investigated whether their willingness to have their children vaccinated would be influenced by nonhealthcare sources and what the risk factors were. The demographic information included sex and age of the CALLS, age of the CALLSs’ mother, marital status of the CALLSs’ family, the highest education level of the family, annual family income, and schooling status of the CALLS. Medical records were used to collect disease-related information such as remission and relapse status.

The Working Alliance Inventory—Short Revised (WAI-SR) was used to assess the therapeutic alliance between the parents and the attending oncologist [12]. The WAI-SR was only collected from parents who had been recommended to vaccinate against COVID-19 by the attending oncologist. The scale contains 12 items in 3 dimensions of working alliance: agreement on therapeutic tasks, agreement on therapeutic goals, and development of an affective bond between parents and the therapists. Items are rated on a 7-point Likert scale, from 1 (never) to 7 (always), with an overall score between 12 and 84. Higher scores indicate a higher level of therapeutic alliance between the parents and the attending oncologist. The scale demonstrated acceptable reliability and validity in Chinese population studies involving cancer patients and caregivers [13].

The Chinese version of the “Parent Attitudes about Childhood Vaccines” questionnaire (PACV) was used to assess parental willingness to vaccinate their CALLS in order to eliminate the confounding factor that influences parental willingness to vaccinate their CALLS [14]. The questionnaire comprises 2 items of childhood vaccination behavior, 4 items of safety/efficacy, and 9 items of general attitudes/trust about pediatric vaccinations. The score ranges from 0 (not hesitant) to 2 (hesitant) and was assigned for all 15 answers (total score ranged from 0 to 30). Higher scores indicate greater hesitancy, while lower scores indicate greater acceptance of non-COVID vaccines.

### 2.3. Statistical Analysis

In the first part, a multivariable binary logistic regression analysis was carried out to find the factors most related to the family willingness to vaccinate their CALLS. Then, the willingness in CALLS parents who received and who did not receive recommendations from their oncologists were compared. To alleviate potential bias between the two comparison groups, participants were matched by propensity scores to reach head-to-head comparison as further validation to multivariable analysis. A McNemar’s test was applied to assess the difference of willingness between the matched samples. A paired-t test was used to compare the difference of PACV scores between the matched groups.

In the second part (subgroup analysis), parents who received recommendations were divided into two groups based upon the median of the WAI-SR scores, representing the therapeutic alliance between the parents and the attending oncologist, and then the willingness was compared. Likewise, multivariable logistic regression was first used to find independent variables associated with the willingness, and then the result with a propensity score-matched (PSM) comparison was conducted for further validation, including the McNemar’s test for the willingness and paired-t test for PACV scores. 

PSM aims to balance uncontrolled baseline variables in real-world settings that incite bias to statistical test results. Propensity scores were calculated with all baseline variables in regression models. A greedy nearest neighbor matching method was adopted to match participants by such scores. As such, participants were matched to mimic randomized settings in clinical trials and were referred to as quasirandomization. This method may reduce selection bias, which is intrinsic to real-world study. PSM analyses in the study were carried out by means of a conditional logistic regression model with a caliper width of 0.04. To assess the matching performance, the standardized difference was calculated for each of the matched variables. According to Austin PC et al., a standardized difference of over (√((n1 + n2)/n1∗n2))∗1.96 is regarded as imbalanced matching (n1 and n2 represent the corresponding sample size of the unmatched sample) [15].

Continuous variables were expressed as means with standard deviation (SD), and categorical variables were expressed in frequencies and percentages. The significant threshold was defined at the value of *p* < 0.05 for paired tests. The nearest matching model was used in the statistical analysis for propensity score matching, which was carried out using the R extensions of the SPSS V.24.0 program. Other statistical analyses, including logistic regression, McNemar’s test, and paired-t test, were applied in the SPSS V.26.0 program.

## 3. Results

### 3.1. Demographic and Oncological Information

A total of 439 CALLS families agreed to participate and were enrolled in the study, with 15 parents being excluded because they did not complete the follow-up questionnaires during the clinical interviews. The final analysis included 424 CALLS families. The mean remission age of the CALLS was 5.99 ± 3.40 years old (median 5), including 230 males and 194 females, and the mean age of the mother of the CALLS at remission was 33.18 ± 4.07 (median = 32) years. Ninety-two children (21.7%) were at the post-treatment surveillance phase (less than 1 year of remission) and 146 (34.4%) children were on remission for 3 years or more. Over one third of the children (165, 38.9%) had a history of relapse. Sixty-one families (14.4%) were divorced, and only fifty-one families (12.0%) had bachelor’s degrees or higher. Thirty-three families (7.8%) had low family income (less than CNY 20,000 per year). A total of 191 families (45.0%) were preparing for school return and the other 233 families (55.0%) were not currently preparing for school. Table 1 shows the demographics and disease details of the included families.

During the interview with the oncologists, 91 families (21.4%) agreed to vaccinate the children 1 year after remission, 165 families (38.9%) disagreed to vaccinate the children, and the other 168 families (39.6%) were hesitant to vaccinate with the COVID-19 vaccines. The most common reasons against vaccination were “lack of recommendations from professional personnel” (n = 84, 50.9%), followed by “belief in potential serious consequences on the diseases” (n = 51, 30.9%), “no need to vaccinate” (n = 20, 12.1%), and others (n = 10, 6.0%) (Appendix A). As for other vaccines, the mean PACV score was 2.93 ± 2.08 (median 3). A total of 175 families (41.2%) reported that advice from nonhealthcare sources would influence their decision to vaccinate their children (Appendix A). The most prevalent source of information was the internet (n = 78, 44.6%), followed by people close to them (n = 52, 29.1%), rumors about side effects (n = 32, 18.3%), and others (n = 13, 7.4%).

### 3.2. Oncologist Recommendation Increases COVID-19 Vaccination Willingness

A total of 118 parents (27.8%) received detailed recommendations (recommendation group, RG) from their treating oncologists, and the other 306 parents (72.2%) did not receive the recommendation (control group, CG). In the multivariable logistic regression analysis, only oncologists’ recommendation was found to significantly predict the willingness to receive vaccinations (OR = 3.17, *p* < 0.001, see Table 2). To validate the result, the recommendation group was propensity score-matched with the control group. The variables to enter the regression model of matching included the following: age at remission among CALLS and their mother, sex, time since remission, school preparation, cancer relapse, parent education, family income, marital state, and nonhealthcare information influence. A total of 114 pairs from the two groups were successfully matched. According to the formula (√((n1 + n2)/n1∗n2))∗1.96, the largest imbalance limit was 0.21. The results showed that the standardized mean difference (SMD) of each variable was less than 0.21, and the detailed balance tests of each variable, as well as the summary of SMD before and after matching, are shown in Appendix A. There were 42/114 parents who agreed to COVID-19 vaccination in the RG, as compared to 18/114 parents in the CG. The McNemar test revealed a significant difference between the two groups (*p* < 0.001, see Figure 1A). The PACV score was compared between the two groups to compare the difference in attitude toward non-COVID vaccines. The mean PACV score in the RG was 2.77 ± 2.02, and the mean PACV score in the CG was 2.75 ± 1.96. By comparison, there was no significant difference in PACV scores (t = −0.065, *p* = 0.95) between the two groups (Figure 1B).

### 3.3. Parent–Oncologist Alliance Affects Recommendation Strength

Next, we investigated the relationship between the patient–oncologist alliance and the willingness to vaccinate in the RG (n = 118). Multivariable analysis showed that only WAI-SR scores were found to predict the acceptance for COVID-19 vaccine (*p* < 0.001, see Table 3). The mean score of WAI-SR in the RG was 43.78 ± 14.80 (median score of 42). Then, parents with a score of more than 42 (alliance group, AG, n = 56) were propensity score-matched with parents with a score of 42 or lower (nonalliance group, NAG, n = 62). The variables to enter the regression model of matching were in line with the PSM analysis among the RG and CG.

There was a total of 32 pairs that were successfully matched from the two groups. The largest imbalance limit was 0.36. The results showed that the SMD of each variable was less than 0.36, and the detailed balance tests of each variable and the summary of SMD before and after matching are shown in Appendix A. There were 21 (65.6%) parents in the AG and 5 (15.6%) parents in the NAG. The McNemar’s test revealed a statistically significant difference between the two groups (*p* < 0.001, see Figure 2A). The PACV scores of the two groups were compared to determine the difference in attitude toward non-COVID vaccines. The mean PACV score in the AG was 3.22 ± 2.37, and the mean PACV score in the NAG was 1.97 ± 1.61. By comparison, there was a significant difference in PACV scores (t = 2.328, *p* < 0.05) between the two groups (Figure 2B).

## 4. Discussion

In the current study of 424 CALLS parents in one single-center location, risk factors related to vaccination acceptance were analyzed. By multivariable analysis, a detailed recommendation from a trusted oncologist was the independent variable. This was further validated by PSM comparison. The same method was used in the subgroup analysis, and the patient–oncologist alliance was found to be associated with the willingness to vaccinate. Because few studies to date have evaluated the safety or efficacy of inactivated COVID-19 vaccination in pediatric patients with cancers, this study provided preliminary data on the attitudes toward COVID-19 vaccination of the pediatric hematologic cancer population, which is rarely studied.

The pediatric population to vaccinate in the study was a sensitive topic; parents were more likely to report an intention to vaccinate themselves than their child, with less concern about the side effects for themselves than for their child [16,17,18,19]. The situation would be more complicated in pediatric patients with solid cancer or hematologic malignancies due to the prevalence of chronic, comorbid health conditions and compromised immune status [20]. Even after treatment completion, all patients in the study contacted the medical service on a regular basis to inquire about home nursing, nutrition shortages, and other psychiatric burdens. The majority of CALLS are home-bound or even home-restrained against the public [21]. Unlike other vaccines administered during the early stages of child development, parents had complex beliefs or concerns about COVID-19 vaccination. In the current study, we found only 21.4% of all parents were willing to accept COVID-19 vaccination in the CALLS, which was significantly lower than that found in English-speaking countries [19,20]. However, the PACV scores for non-COVID vaccines were relatively low (median = 3), suggesting a high level of acceptance of non-COVID vaccines. The gap in the attitude between COVID-19 and other vaccines was probably due to the validated safety profiles of traditional vaccines.

Our findings supported the prior hypothesis that parental acceptance of vaccination for children with cancer could be influenced by oncologist recommendations [22]. Furthermore, the importance of physician consultation suggests that parent education by the oncologist may help change the attitude of those parents who would have disagreed to vaccinate the patient [22]. During the consultation process, the risks and benefits of vaccination should be thoroughly discussed, and questions about long-term effects on disease prognosis of the individual patients should be addressed based on immunology system function [21].

The study also emphasizes the significance of parent–oncologist alliance in vaccination acceptance. This is consistent with our previous findings that patient–oncologist alliance was positively associated with higher treatment adherence [23]. The therapeutic alliance, a modifiable factor that connects a patient to their medical provider, has been linked to improved social function, mental health, and overall health-related quality of life [24]. As a result, establishing a strong parent–oncologist alliance may be a feasible way to increase the willingness of COVID-19 vaccination, and the quality of oncologist communication skills and patient confidence regarding the oncologist may be the critical influencing factors [24]. In addition to oncologists, parents also obtain information from nonhealthcare resources, with the internet being the most common source, according to the current study. Although this result was in line with previous reports of vaccination hesitancy in the general public, the result in the present study should be discussed cautiously, as the population was CALLS parents, and compared to the healthy people, rumors on the internet about adverse events could drastically sway the attitude of CALLS parents [25].

The current study has several limitations. The sample recruitment of CALLS defined that the parents’ willingness only applied to the hematology cancers and not to solid cancers. Furthermore, this is a single-center study, and the findings may not be applicable in English-speaking countries or other areas of China due to differences in therapeutic protocols and sociolect-demographic variables. The cross-sectional nature of the data suggests the attitude of the parents at a limited time frame, which encourages further longitudinal follow-up to see if attitudes change over time in CALLS parents.

## 5. Conclusions

Parents were generally unwilling to vaccinate children with acute lymphoblastic leukemia because of a lack of professional, trusted sources of consultation. Oncologist recommendations would significantly fill the knowledge gap and increase the willingness. A stronger alliance between therapists and parents appears to increase parents’ trust. Nevertheless, the conclusions in the present study should be generalized with caution due to the nature of a single-center study in a tertiary hospital of north China. Further multiple-center longitudinal study involving patients with different therapeutic backgrounds should be conducted to validate our findings.

## Figures and Tables

**Figure 1 vaccines-10-02016-f001:**
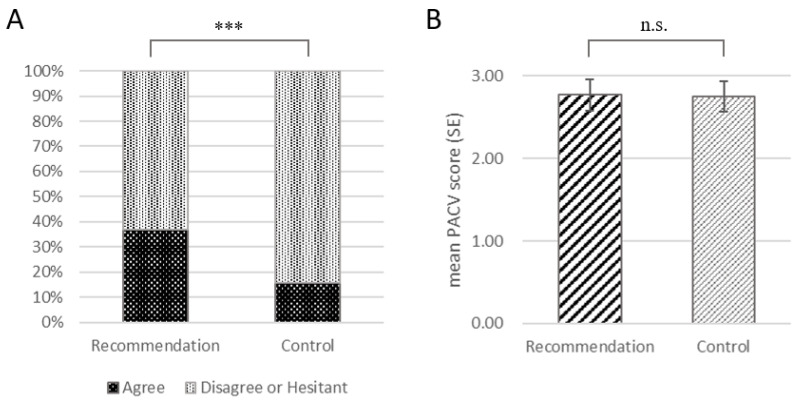
The comparison of COVID-19 vaccination acceptance and PACV scores between the recommendation group and control group (both in matched samples). (**A**), the willingness for vaccination in the recommendation group was significantly higher than in controls (*p* < 0.001); (**B**), there was no significant difference in the mean PACV scores between the two groups. Notes: the “recommendation group” means parents who were recommended by the oncologists, and the “control group” means parents who did not receive recommendation from the oncologists. (*** *p* < 0.001; n.s.: no significant difference).

**Figure 2 vaccines-10-02016-f002:**
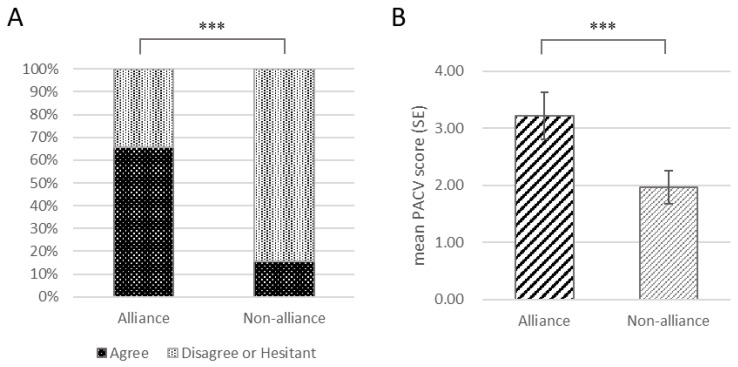
The comparison of parental wiliness against COVID-19 vaccine and non-COVID vaccines between the alliance group and nonalliance group (both in matched samples). (**A**), the willingness to vaccination in the alliance group was significantly higher than in the nonalliance group (*p* < 0.001); (**B**), there was a significant difference in the mean “Parent Attitudes about Childhood Vaccines” questionnaire (PACV) scores between the two groups (*** *p* <0.001).

**Table 1 vaccines-10-02016-t001:** Baseline characteristics of the included CALLS families.

Factor		All CALLS Families	Agree N = 91 (21.4%)	Disagree N = 165 (38.9%)	Hesitant N = 168 (39.6%)
Child Characteristics					
Sex	Female	194 (45.8%)	34 (37.4%)	84 (50.9%)	76 (45.2%)
	Male	230 (54.2%)	57 (62.6%)	81 (49.1%)	92 (54.8%)
Age at Remission	Mean ± SD	5.99 ± 3.40	6.31 ± 3.94	6.05 ± 2.83	5.76 ± 3.60
Age at Remission (Category)	<6	220 (51.9%)	50 (54.9%)	75 (45.5%)	95 (56.5%)
	≥6	204 (48.1%)	41 (45.1%)	90 (54.5%)	73 (43.5%)
Remission Time (Year)	<1	92 (21.7%)	20 (22.0%)	34 (20.6%)	38 (22.6%)
	1–2	78 (18.4%)	15 (16.5%)	32 (19.4%)	31 (18.5%)
	2–3	108 (25.5%)	30 (33.0%)	42 (25.5%)	36 (21.4%)
	≥3	146 (34.4%)	26 (28.6%)	57 (34.5%)	63 (37.5%)
School Preparation	Yes	191 (45.0%)	46 (50.5%)	73 (44.2%)	72 (42.9%)
	Not yet	233 (55.0%)	45 (49.5%)	92 (55.8%)	96 (57.1%)
Relapse History	Ever	165 (38.9%)	30 (33.0%)	68 (41.2%)	67 (39.9%)
	Never	259 (61.1%)	61 (67.0%)	97 (58.8%)	101 (60.1%)
Family Characteristics					
Age of Mother at Remission	Mean ± SD	32.50 ± 3.87	32.81 ± 4.04	32.58 ± 3.81	32.26 ± 3.84
Age of Mother at Remission (Category)	<33	227 (53.5%)	43 (47.3%)	89 (53.9%)	95 (56.5%)
	≥33	197 (46.5%)	48 (52.7%)	76 (46.1%)	73 (43.5%)
Education Level ^†^	High school or below	224 (52.8%)	45 (49.5%)	83 (50.3%)	96 (57.1%)
	Junior college	149 (35.1%)	37 (40.7%)	53 (32.1%)	59 (35.1%)
	Bachelors or above	51 (12.0%)	9 (9.9%)	29 (17.6%)	13 (7.7%)
Annual Family Income, CNY ^‡^	<20,000	33 (7.8%)	7 (7.7%)	14 (8.5%)	12 (7.1%)
	20,000 to 100,000	168 (39.6%)	39 (42.9%)	64 (38.8%)	65 (38.7%)
	100,000 to 200,000	60 (14.2%)	16 (17.6%)	17 (10.3%)	27 (16.1%)
	>200,000	163 (38.4%)	29 (31.9%)	70 (42.4%)	64 (38.1%)
Marital Status	Married	363 (85.6%)	77 (84.6%)	142 (86.1%)	144 (85.7%)
	Divorced	61 (14.4%)	14 (15.4%)	23 (13.9%)	24 (14.3%)
Oncologist Recommendation	Yes	118 (27.8%)	44 (48.4%)	13 (7.9%)	61 (36.3%)
	No	306 (72.2%)	47 (51.6%)	152 (92.1%)	107 (63.7%)
Swayed by Nonhealthcare Information	Yes	175 (41.3%)	36 (39.6%)	78 (47.3%)	61 (36.3%)
	No	249 (58.7%)	55 (60.4%)	87 (52.7%)	107 (63.7%)

CALLS: childhood acute lymphoblastic leukemia survivor; ^†^: highest education level of the family; ^‡^ based on the exchange rate of CNY against USD in 2021, CNY 20,000 in the table = USD 2824, CNY 100,000 = USD 14,120, CNY 200,000 = USD 28,240. Categorical variables were illustrated as numbers and percentage, and continuous variables were illustrated as mean ± standard deviations (SD).

**Table 2 vaccines-10-02016-t002:** Multivariable logistic regression of vaccine acceptance in CALLS families.

Variables (Reference)	*β*	Standard Error	Wald	*p*	OR	95% CI
Sex (Female)	0.36	0.26	1.97	0.16	1.43	0.87–2.36
Age at Remission (Continuous)	0.02	0.04	0.16	0.69	1.02	0.94–1.09
Remission Time (Continuous)	−0.03	0.11	0.10	0.76	0.97	0.78–1.19
School Preparation (Not yet)	0.17	0.25	0.46	0.50	1.18	0.73–1.93
Relapse History (Never)	−0.36	0.26	1.87	0.17	0.70	0.42–1.17
Mother’s Gestating Age (Continuous) ^a^	0.00	0.05	0.00	0.97	1.00	0.90–1.11
Education Level (High school or below) ^b^	0.15	0.25	0.34	0.56	1.16	0.71–1.88
Family Income (Poverty) ^c^	0.07	0.46	0.03	0.87	1.08	0.43–2.68
Marital Status (Divorced)	0.00	0.35	0.00	1.00	1.00	0.51–1.98
Swayed by Nonhealthcare Information (No)	−0.06	0.25	0.05	0.82	0.94	0.58–1.55
Recommendation (No)	1.15	0.25	20.94	<0.001	3.17	1.93–5.20
Constant	−2.05	1.55	1.76	0.19	0.13	

CALLS: childhood acute lymphoblastic leukemia survivor; OR: odds ratio; 95% CI: 95% confidence interval; ^a^: calculated as mother’s age minus the age of the CALLS; ^b^: highest education level of the family, which was transformed into two groups (high school or below vs. junior college and bachelor’s or above); ^c^: annual income of the family, which was transformed into two groups (poverty: <20,000, nonpoverty: ≥20,000).

**Table 3 vaccines-10-02016-t003:** Multivariable logistic regression of vaccine acceptance in CALLS parents recommended to receive vaccination.

Variables (Reference)	*β*	Standard Error	Wald	*p*	OR	95% CI
Sex (Female)	0.59	0.55	1.14	0.29	1.80	0.61–5.32
Age at Remission (Continuous)	0.12	0.08	2.32	0.13	1.13	0.97–1.32
Remission Time (Continuous)	−0.04	0.22	0.03	0.85	0.96	0.62–1.48
School Preparation (Not yet)	0.43	0.54	0.61	0.43	1.53	0.53–4.45
Relapse History (Never)	0.38	0.56	0.46	0.50	1.46	0.48–4.43
Mother’s Gestating Age (Continuous) ^a^	0.02	0.11	0.05	0.82	1.03	0.83–1.27
Education Level (High school or below) ^b^	−0.31	0.53	0.35	0.55	0.73	0.26–2.06
Family Income (Poverty) ^c^	−0.44	1.14	0.15	0.70	0.65	0.07–6.03
Marital Status (Divorced)	−0.12	0.72	0.03	0.86	0.88	0.21–3.65
Swayed by Nonhealthcare Information (No)	−0.62	0.56	1.21	0.27	0.54	0.18–1.62
WAI-SR (Continuous)	0.14	0.03	26.34	<0.001	1.15	1.09–1.21
Constant	−7.87	3.61	4.75	0.03	0.00	

CALLS: childhood acute lymphoblastic leukemia survivor; OR: odds ratio; 95% CI: 95% confidence interval; WAI-SR: Working Alliance Inventory—Short Revised; ^a^: calculated as mother’s age minus the age of the CALLS; ^b^: highest education level of the family, which was transformed into two groups (high school or below vs. junior college and bachelor’s or above); ^c^: annual income of the family, which was transformed into two groups (poverty: <20,000, nonpoverty: ≥20,000).

## Data Availability

Data are available on reasonable request. The personal information of the study participants is kept strictly confidential as mandated by the request of the study participants.

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
