# Peer review of "Parent Acceptance toward Inactivated COVID-19 Vaccination in Children with Acute Lymphoblastic Leukemia: The Power of Oncologist and Alliance"

_vaccines, 2022, doi:10.3390/vaccines10122016_

Round 1

Reviewer 1 Report

REVIEWER'S REPORT

Manucsript title: Parent Acceptance toward Inactivated Covid-19 Vaccination in Children with Acute Lymphoblastic Leukemia: The Power of Oncologist and Alliance  (Authors:

 Yifei Ma , Nianqi Liu, Guanqing Zhong, Dao Wang, Lu Cao, Shenrui Bai, Pengfei Zhu, Ao Zhang, Xinjia 5Wang ). The study outlined in this manuscript aims to identify factors associated with vaccine acceptance and survey parents' willingness to vaccinate their children with acute lymphoblastic leukemia (CALLS). The study included 424 families in total. Validation was accomplished by using propensity score-matched (PSM) analysis. The majority of respondents has been revealed to be unsure or disagree about vaccination, with only around one-fifth agreeing.

   To be honest, I am not a supporter of such articles and somewhat skeptical of child vaccination because the long-term effects of vaccination are unknown.On the other hand, who knows, maybe other readers of this article will recognize the significance of this study. This article could be accepted with some editing and supplements.

    First and foremost, I strongly advise proofreading the WHOLE TEXT of the manuscript and supplementary materials, as this will have a significant impact on the paper's quality.

    In Abstract (page 1): the first sentence (line 27) "The current study seeks to survey..." should be corrected as "The current study aims to survey..."; the sentence (in line 30) "...were recruited into interviews..." should be written "...were recruited for interviews..."; the sentences (in lines 31-34)  "Oncologist recommendation and parent-oncologist alliance were assessed for association with acceptance status after adjusting for socio-economic confounders. Propensity score-matched (PSM) analysis was applied for validation" should be paraphrased as "After controlling for socioeconomic factors, the association of oncologist recommendation and parent-oncologist alliance with acceptance status was investigated. For validation, propensity score-matched (PSM) analysis was used."; in line 37 "...and non-healthcare resources hindering parents from vaccination, respectively." must be corrected as " were the most common reasons and non-healthcare resources that kept parents from getting vaccinated, respectively "; in line 40 "...which proved by PSM comparison..." should be replaced by "...as demontrated by PSM comparison..."; in sentence (line 41) "Exploratory analysis disclosed " should be corrected as "An exploratory analysis revealed..."; The sentences (in lines 43-49) should be written as " Due to a lack of professional recommendation resources and the potential for serious consequences, parents were generally reluctant to vaccinate their CALLS. The recommendation of oncologists, which was influenced by the parent-oncologist alliance, significantly increased acceptance. This study emphasizes the critical role of oncologists in vaccinating cancer survivors and can be used to promote COVID-19 vaccines among vulnerable populations."

   In Introduction (page 2): the first sentence (lines 54-55) "...has been evidenced to be catastrophic to patients with cancers, and pediatric patients with hematologic malignancies proved the most vulnerable to such attacks." can be paraphrased as "...has been shown to be disastrous for cancer patients, with pediatric patients with hematologic malignancies being the most vulnerable to such attacks."; in line 57 "...across the globe." should be replaced by "...worldwide"; in lines 61-65, the sentence "Although previous research has proved that children following allogeneic hematopoietic stem cell transplant are able to mount protective immune responses to inactivated influenza vaccination[6], there is a paucity of research regarding coronavirus disease 2019 (COVID-19) vaccination in children with hematological malignancies[7]." should be paraphrased as " Although previous research has shown that children who have had allogeneic hematopoietic stem cell transplants can mount protective immune responses to inactivated influenza vaccination[6, 7], there has been little research on corona-virus disease 2019 (COVID-19) vaccination in children with hematological malignancies. "; the sentence in line 66 "...it is imperative to implement vaccination to CALLS when appropriate and available." should be replaced by "...it is critical to provide CALLS with vaccination when appropriate and available."; the sentence in lines 68-69 "Literature showed relatively high acceptance of vaccination for parents of healthy children, and the research of hesitancy for parents of childhood cancer survivors, on the other hand, has been relatively lacking." sould be paraphrased as "The literature revealed relatively high acceptance of vaccination for parents of healthy children, whereas research on hesitancy for parents of childhood cancer survivors was relatively lacking."; in line 70 "...still lacking..." should be replaced by "...still scarce..." in lines 71-72 "...demonstrated the general reluctance thanks to uncertainties on the safety and efficacy of COVID-19 vaccination [8]." should be replaced by "...revealed widespread apprehension due to concerns about the safety and efficacy of COVID-19 vaccination [8]."; in line 72 "Considering..." should be replaced by "Given..."; in line 73 "...in the time of cancer remission phase and high rate vaccination..." should be replaced by "... during the cancer remission as well as the high rate..."; in line 74 "... the data on the parents’ attitude of COVID-19 vaccination may help understand how to increase vaccination coverage [5]." should be corrected as "... data on parents' attitudes toward COVID-19 vaccination may aid in understanding how to increase vaccination coverage[5].";  in lines 78-79 "... may receive vaccination regardless of chemotherapy status[9]." should be replaced by "... regardless of chemotherapy status, could receive vaccination [9]."; in line 81 "Parent education..." should be replaced by "Parental education..."; in line 82 " ... was shown to significantly alter the vaccine hesitancy[9]." should be replaced by "...has been shown to significantly reduce vaccine apprehension [9]."; in line 83 "...in Chine showed..." should be replaced by "...in  China found that..."; in lines 83-84 "...expressed hesitancy or refusal to vaccinate, and the most prevalent reason was lack of professional..." should be written as "...were hesitant or refused to vaccinate, with a most common reason being a lack of professional..."; in lines 85-87 "...notably, the person who would most influence decisions of patients was the doctor in charge of 86 treatment[10]." should be replaced by "Notably, the person most likely to influence patient decisions was the doctor in charge of treatment[10]. "; in line 88, the first part of sentence "Although the oncologists of the childhood patients’ families..." should be written as "Although childhood cancer patients' oncologists..."; in lines 91-92, the sentence "Parents’ adherence to child therapy has been shown to be affected by the therapeutic alliance with the oncologists, which represents the level of mutual trust and understanding on both sides of health care[11]." should replaced by " It has been demonstrated that the therapeutic alliance with the oncologists, which stands for the degree of mutual trust and understanding on both sides of health care, has an impact on parents' adherence to child therapy."; in line 94, "...recommendation"  have to be written a "...recommendations..."; in lines 99 "In the current study, we seek to find independent variables affecting..." should be replaced by "The current study aims to identify independent variables influencing...", and in lines 100-101 "...and then interrogate how patient-oncologist alliance mediates the association between recommendation and parents’ attitude." should be replaced by "...and then investigate how the patient-oncologist alliance mediates the relationship between recommendation and parental attitude."

In Materials and Methods (page 3) sub-section Participant Enrollment. In lines 112-114, the sentences "Participants must give oral or written informed consent to participate in the study before enrollment. The study was approved by the institutional review board of the Second Affiliated Hospital of Shantou University Medical College." I recommend to pharaphrase as "Prior to enrollment, participants must provide either verbal or written informed consent to participate in the study. The study was approved by the institutional review board of Shantou University Medical College's Second Affiliated Hospital."

In sub-section Patient Follow-up and Instrument Evaluation. The sentence in lines 117-120 "Parents of the CALLS were semi-structured interviewed during the outpatient visit to assess their family willingness of the COVID-19 vaccine for the CALLS one year after remission, the recommendation status towards the vaccination by the attending oncologist, and the non-healthcare sources that could sway their attitude about the vaccination." shoud paraphrased as "During the outpatient visit, parents of the CALLS underwent semi-structured interviews to determine whether their family would be willing to receive the COVID-19 vaccine for the CALLS one year after remission, the attending oncologist's recommendation status regarding the vaccination, and any outside influences that might affect their attitude toward the vaccination."; in line 121 "If parents say they were unsure about the attitude of the doctor towards the vaccination,..." should be replaced by" If parents expressed concerns about the doctor's stance on vaccination,..."; in line 123 "...were blind to..." should be replaced by "...were to aware of..."; in line 126 "...so as to divide the attitudes into..." should be replaced by "...so attitudes were divided into..."; the sentence in lines 130-131 "In addition, we assessed whether their willingness to have their children vaccinated would be swayed by other non-healthcare sources and what the risk factors were." should be paraphrased by "Furthermore, we investigated whether their willingness to have their children vaccinated would be influenced by non-healthcare sources, as well as the risk factors."; in  134-135 "The disease-related details 134 included remission and relapse status were collected from medical records." shoud be replaced by "Medical records were used to collect disease-related information such as remission and relapse status."; in 136-137 "The therapeutic alliance between the parents and the attending oncologist was evaluated by the Working Alliance Inventory-Short Revised (WAI-SR)[12]" should be replaced by "The Working Alliance Inventory-Short Revised (WAI-SR) was used to assess the therapeutic alliance between the parents and the attending oncologist [12]."; the sentence in lines 142-144 "The scale proved satisfactory reliability and validity in the Chinese population studies associated with cancers patients and their caregivers[13]. " should be paraphrased as "The scale demonstrated acceptable reliability and validity in Chinese population studies involving cancer patients and caregivers [13]."; in lines 145-147 "To eliminate the confounding factor that influences parental willingness to vaccinate their CALLS, the Chinese version of “Parent Attitudes about Childhood Vaccines” questionnaire (PACV) was applied to assess the parental wiliness against non-COVID vaccines[14]." The Chinese version of the "Parent Attitudes about Childhood Vaccines" questionnaire (PACV) was used to assess parental willingness to vaccinate their CALLS in order to eliminate the confounding factor that influences parental willingness to vaccinate their CALLS [14]."; the sentence in lines 151-152 " Higher scores indicate a higher level of hesitancy, and lower scores indicate a higher level of acceptance of non-COVID vaccines." could be paraphrased as "Higher scores indicate greater hesitancy, while lower scores indicate greater acceptance of non-COVID vaccines."

In sub-section Statistical Analysis. The sentence In lines 174-176 " The statistical analysis used in propensity score matching applied the nearest matching model and was performed in the R extensions of SPSS V.24.0 software" should be replaced by "The nearest matching model was used in the statistical analysis for propensity score matching, which was carried out using the R extensions of the SPSS V.24.0 program."

In Results, sub-section Demographic and Oncological Information. The sentence in lines 179-180 "...families consented to participate and were enrolled in the study, of whom 15 parents did not fulfill the follow-up questionnaires during the clinical 180 interviews and thus were excluded." should be rewritten as "...families agreed to participate and were enrolled in the study, with 15 parents being excluded because they did not complete the follow-up questionnaires during the clinical interviews."; in lines 186-187 "... had bachelors’ degrees or above." should be repaced by " should be corrected as "... had bachelors’ degrees or higher."; in lines 190-191, the sentence " The demographics and disease details of the included families were shown in Table 1." should be replaced by " Table 1 shows the demographics and disease details of the included families."; in lines 199-200 "...reported that the decision to vaccinate the children would be swayed by advice from non-healthcare sources" should be replaced by "... reported that advice from non-healthcare sources would influence their decision to vaccinate their children."

 In sub-section  Oncologist Recommendation Increases COVID-19 Vaccination Willingness (page 5 ). Iin lines 2011-2012 "... willingness to vaccinate" I recommend to replace by "... propensity to receive vaccinations"; in lines 216-217 "... There was a total of 114 pairs that were successfully matched from the two groups" should be rewritten as "A total of 114 pairs from the two groups were successfully matched."; in lines 219-220 " and the detailed balance tests of each variable and the summary of SMD before and after matching..." should be relaced by "...as well as the summary of SMD before and after matching, were shown in Supplementary Table 1 and Supplementary Figure 2.' in lines 223-224, the sentence " By the McNemar test, there was a significant difference between the two groups (p<0.001, see Figure 1A)." should be rewritten  as " The McNemar test revealed a significant difference between the two groups (p< 0.001, see Figure 1A).".

In sub-section Parent-oncologist Alliance Affects Recommendation Strength (page 7). In line 245 "...parents who scored over..." should be written as  "... parents with a score of 42 or higher..."; in line 246  "...parents who scored 42 or less..." should be written as "... parents with a score of 42 or lower..."; in line 254 " By the McNemar test, there was a significant" must be written as " The McNemar test revealed a statistically significant difference..."; in lines 255-256, the sentence " The PACV scores were compared between the two groups to compare the difference in attitude towards non-COVID vaccines." should be written as " The PACV scores of the two groups were compared to determine the difference in attitude toward non-COVID vaccines."

In Discussion (page 8). In lines 274-275 "... associated with vaccination..." should be written as "...related to vaccination..."; the sentence in lines 277-278 " In the subgroup analysis, the same method was applied and the patient-oncologist alliance was found to be associated with the willingness to vaccinate" should be written as " The same method was used in the subgroup analysis, and the patient-oncologist alliance was found to be associated with the willingness to vaccinate."; in line 279, instead of "As..." you must write "Because..."; in lines 280-281 " the part of sentence "...this study gave preliminary data on the attitudes over the COVID-19 vaccination of the pediatric hematologic cancer population who are seldom studied" must be rewritten as "... this study provided preliminary data on the attitudes toward COVID-19 vaccination of the pediatric hematologic cancer population, which is rarely studied."; in line 283 " The pediatric population to vaccinate in the study was one of the sensitive topics" must be written as " One of the sensitive topics in the study was the pediatric population to vaccinate; the sentence in lines 285-287 " When it comes to pediatric patients with solid cancer or hematologic malignancies, the situation would be more complicated due to the prevalence of chronic, comorbid health conditions and compromised immune status[20]." could be rewitten as  " The situation would be more complicated in pediatric patients with solid cancer or hematologic malignancies due to the prevalence of chronic, comorbid health conditions and compromised immune status[20].";  in lines 288-289 " ...all patients in the 288 study were in frequent contact..." could be written as "...all patients in the study contacted the medical service on a regular basis..."; in line 290 " Most CALLS become home-bound..." should be written " The majority of CALLS are home-bound..."; in line 291-293, the sentence " As different from other vaccines which are done during the early development milestones, parents subsequently held complicated beliefs or concerns about COVID-19 vaccination." must be rewritten as " Unlike other vaccines administered during the early stages of development, parents had complex beliefs or concerns about COVID-19 vaccination."; the sentence in lines 300-302 should replaced by " Our study proved the prior hypothesis that parent acceptancy of vaccination on 300 childhood cancer patients could be influenced by oncologist recommendations[22]" should be replaced by " Our findings supported the prior hypothesis that parental acceptance of vaccination for children with cancer could be influenced by oncologist recommendations. "; in line 304 " In the process of consultation" must be replaced by "During the consultation process..."; and in line 305 "... should be discussed properly, and questions on long-term effects on disease 305 prognosis of the individual patients should be addressed by means of immunology system function as well[21]" could be corrected as "...should be thoroughly discussed and questions about long-term effects on disease prognosis of the individual patients should be addressed by using of immunology system function.": in line 308 "...highlights the importance..." should be replaced by "... emphasizes the significance..."; in line 309

 " This is in line with our past findings..." replace by " This is consistent with our previous findings..."; in line 310 " associated with greater adherence of treatment adherence..." should be replaced by " associated with higher treatment adherence..."; in line 311, instead of "...bonds..." write "...connects..."; in lines 311-312, instead of " ...has been shown to predict better social function, mental and general health-related quality of life" write "... has been linked to improved social function, mental health, and overall health-related quality of life [24]."; in lines 313-314 " Therefore, the establishment of a strong parent-oncologist alliance could be a practicable way..." should be rewritten as " As a result, establishing a strong parent-oncologist alliance may be a feasible way..."; in lines 317-318  " and the majority source of the information comes from the internet, as the present study suggests. " could be replaced by " ...with the internet being the most common source, according to the current study." in line 321 "...rumors on the internet on adverse events" write as " rumors on the internet about adverse events..."; in lines 325-327, the sentence should be written " Furthermore, this is a single-center study, and the findings may not be applicable in English-speaking countries or other areas of China due to differences in therapeutic protocols and sociolect-demographic variables."; in line 328-329 " ...to see changes of attitude in parents of CALLS." should be written like "... to see if attitudes change in CALLS parents."

The sentence in lines 334-335 " A better alliance between therapists and parents seems to enhance the parents’ trust..." should be rewritten as " A stronger alliance between therapists and parents appears to increase parents' trust..."

In supplementary materials. In page 2, Figure 2, instead of "efficacy evaluation of propensity score matching by oncologist recommendation" should be written " Propensity score matching efficacy evaluation based on oncologist recommendation";  instead of "Gestating age of mother:." should be written " Mother's Gestating Age:..";.   

In my opinion, statistical analysis sub-section (page 4) should include a more detailed description of statistics, including references to all statistical methods used in this work, as well as a brief justification of why they are used and how they are superior to and/or more reliable than other/alternative statistical techniques.

The statistical samples were taken from one or more local areas, as the authors briefly stated at the end of the article (lines 325-327), and they are unlikely to reflect the general population (both in China and in other countries). In my opinion, it should be stated in the abstract and/or conclusions.

Author Response

Point-by-point response to reviewer 1:

We are grateful for the review on grammar and paraphrasing and we believe that these suggestions made by the reviewers were constructive to a better manuscript. Nearly all review comments were accepted totally and we appreciate the verbatim paraphrasing suggestions (all revisions were marked in red). However, there are a few comments that require specific clarification as follows. In addition, the last 2 response was to the data analysis protocols.

  1. in line 37 "...and non-healthcare resources hindering parents from vaccination, respectively." must be corrected as " were the most common reasonsand non-healthcare resources that kept parents from getting vaccinated, respectively "

Answer: Thank you for your comments. We did not make it quite clear, and so the original sentence should be changed. “Lack of recommendations from professional personnel (84/165, 50.9%) and massive internet information (78/175, 44.6%) were the most common reason and non-healthcare resources hindering parents from vaccination, respectively.” Revision: “The most common reason that kept parents from getting vaccinated was lack of recommendations from professional personnel (84/165, 50.9%), and massive internet information (78/175, 44.6%) was the main non-healthcare resource against vaccination.” (in line 36-39)

  1. in lines 71-72 "...demonstrated the general reluctance thanks to uncertainties on the safety and efficacy of COVID-19 vaccination [8]." should be replaced by "...revealed widespread apprehension due to concerns about the safety and efficacy of COVID-19 vaccination [8]."

Answer: Thank you for your suggestion. “Vaccine hesitancy” is a common expression in the literaturr and I replaced the “apprehension” in your review suggestion with “hesitancy. Revision: “...revealed widespread hesitancy due to concerns about the safety and efficacy of COVID-19 vaccination[8]”(in line 72-73)

  1. in line 82 " ...was shown to significantly alter the vaccine hesitancy[9]." should be replaced by "...has been shown to significantly reduce vaccine apprehension [9]."

Answer: Thanks for your comments. For the same reason as the previous one, I replaced the “apprehension” in the review opinion with “hesitancy”. Revision: “has been shown to significantly reduce vaccine hesitancy[9]” (in line 83)

  1. In Materials and Methods (page 3) sub-section Participant Enrollment. In lines 112-114, the sentences "Participants must give oral or written informed consent to participate in the study before enrollment. The study was approved by the institutional review board of the Second Affiliated Hospital of Shantou University Medical College." I recommend to pharaphrase as "Prior to enrollment, participants must provide either verbal or written informed consent to participate in the study. The study was approved by the institutional review board of Shantou University Medical College's Second Affiliated Hospital."

Answer: Thanks for your comments. According to the official website of the company, the Second Affiliated Hospital of Shantou University Medical College is the official name. Therefore, in this modification suggestion, we modified the sentence pattern as suggested, but did not modify the name of the hospital. Revision: “Prior to enrollment, participants must provide either verbal or written informed consent to participate in the study. The study was approved by the institutional review board of the Second Affiliated Hospital of Shantou University Medical College.”(in line 114-117)

  1. in line 123 "...were blind to..." should be replaced by "...were to aware of..."

Answer: Thanks for your comments. In this retrospective study, oncologists did not know the specific study plan, so we wrote "be blind to". In order to express our meaning more clearly, we did not modify it as suggested, but added a sentence to explain it. Revision: The attending oncologists of the participants were blind to the research protocol, as we retrospectively interviewed the parents without seeking confirmation from their doctor.” (in line 125-127)

  1. the sentence in lines 130-131 "In addition, we assessed whether their willingness to have their children vaccinated would be swayed by other non-healthcare sources and what the risk factors were." should be paraphrased by "Furthermore, we investigated whether their willingness to have their children vaccinated would be influenced by non-healthcare sources, as well as the risk factors."

Answer: Thanks for your comments. If we change it to "as well as the risk factors", it is not clear what the risk factors are referred to, so the second half of the sentence is unchanged. Revision: “Furthermore, we investigated whether their willingness to have their children vaccinated would be influenced by non-healthcare sources and what the risk factors were.” (in line 134-136)

  1. in lines 211-212 "... willingness to vaccinate" I recommend to replace by "...propensity to receive vaccinations"

Answer: Thanks for your comments. In this sentence, we do not think it is appropriate to replace "willingness" with "propensity" because propensity is easily confused with the propensity score matching analysis used in this article. Revision: In the multi-variate logistic regression analysis, only oncologists’ recommendation was found to significantly predict the willingness to receive vaccinations”. (in line 232-234)

  1. In line 245 "...parents who scored over..." should be written as  "...parents with a score of 42 or higher..."; in line 246  "...parents who scored 42 or less..." should be written as "... parents with a score of 42 or lower..."

Answer: Alliance group was defined as parents with a score of more than 42 (>42, not >=42). Revision: Then, parents with a score of more than 42 (alliance group, AG, n = 56) were propensity score-matched with parents with a score of 42 or lower (non-alliance group, NAG, n = 62).” (in line 269-271)

  1. in line 283 "The pediatric population to vaccinate in the study was one of the sensitive topics" must be written as " One of the sensitive topics in the study was the pediatric population to vaccinate;

Answer: Thanks for your comments. What this means is that childhood vaccination is a sensitive topic in society rather than one of sensitive topics in this study. Revision: The pediatric population to vaccinate in the study was a socially sensitive topic,… ( in line 307)

  1. in line 291-293, the sentence "As different from other vaccines which are done during the early development milestones, parents subsequently held complicated beliefs or concerns about COVID-19 vaccination." must be rewritten as " Unlike other vaccines administered during the early stages of development, parents had complex beliefs or concerns about COVID-19 vaccination."

Answer: Thanks for your comments. To make it clear, we added "children" before "development". Revision: Unlike other vaccines administered during the early stages of children development, parents had complex beliefs or concerns about COVID-19 vaccination. (in line 315-317)

  1. in line 328-329 "...to see changes of attitude in parents of CALLS." should be written like "... to see if attitudes change in CALLS parents."

Answer: Thanks for your comments. Since longitudinal studies are designed to look at changes over time, we added "over time,". Revision: “The cross-sectional nature of the data suggests the attitude of the parents at a limited time frame, which encourages further longitudinal follow-up to see if attitudes change over time in CALLS parents. (in line 351-353)

  1. In supplementary materials. In page 2, Figure 2, instead of "Gestating age of mother:." should be written " Mother's Gestating Age:.."

Answer: Thanks for your comments. The text and figures have all been revised, including Figures 2, 3 and table 1,2 in the supplementary document and Tables 2 and 3 in the main text. (marked in red).

  1. In my opinion, statistical analysis sub-section (page 4) should include a more detailed description of statistics, including references to all statistical methods used in this work, as well as a brief justification of why they are used and how they are superior to and/or more reliable than other/alternative statistical techniques.

Answer: Thanks for your comments. All data analysis steps are described in detail. For example, in “Statistical Analysis”section(in line 159-195),we added “Paired-t test was used to compare the difference of PACV scores between the matched groups.”ï¼› “…including the McNemar’s test for the willingness and Paired-t test for PACV scores.”; and we added other explanation for PSM:“PSM aims to balance uncontrolled baseline variables in real-world settings that incite bias to statistical test results. Propensity scores were calculated with all baseline variables in regression models. A greedy nearest neighbor matching method was adopted to match participants by such scores. As such, participants were matched to mimic randomized settings in clinical trials and were referred as quasi-randomization. This method may reduce selection bias which is intrinsic to real-world study.”; “Other statistical analyses including logistic regression, McNemar’s test, and Paired-t test were applied in the SPSS V.26.0 program.” 

In addition, there are specific variables for PSM inclusion in the text,e.g., “The variables to enter the regression model of matching included the following: age at remission among CALLS and their mother, sex, time since remission, school preparation, cancer relapse, parent education, family income, marital state, and non-healthcare information influence.” (in line 235-238); “The variables to enter the regression model of matching were in line with the PSM analysis among the RG and CG.” (in line 271-272)(Green marks in text). As for why this method is chosen instead of others, it is because the data types only support this analysis method, such as dependent variable as dichotomy variable, logistic regression, paired samples as paired sample test method.

  1. The statistical samples were taken from one or more local areas, as the authors briefly stated at the end of the article (lines 325-327), and they are unlikely to reflect the general population (both in China and in other countries). In my opinion, it should be stated in the abstract and/or conclusions.

Answer: Thanks for your comments. We've already added that to the conclusion. Revision: Nevertheless, the conclusions in the present study should be generalized with caution due to the nature of single-center study in a tertiary hospital of north China. Further multiple-center longitudinal study involving patients with different therapeutic backgrounds should be conducted to validate our findings. (in line 358-361)

Reviewer 2 Report

I have only one observation in the conclusions section:

I would suggest writing the complete acronym for CALLS (children with acute lymphoblastic leukemia) in the conclusions section. This in order to be read continuous.

Conclusions
Parents were generally unwilling to vaccinate CALLS children because…

Parents were generally unwilling to vaccinate children with acute lymphoblastic leukemia because…

It is for author decision

Author Response

Point-by-point response to reviewer:

Thank you very much for the reviewer's suggestion. We changed the CALLS into children with acute lymphoblastic leukemia in the conclusion section, and please see text with gray background in the conclusion section.

"Parents were generally unwilling to vaccinate children with acute lymphoblastic leukemia because of a lack of professional, trusted sources of consultation."

"...parents were generally reluctant to vaccinate their children with acute lymphoblastic leukemia."

Reviewer 3 Report

This manuscript aimed to identify independent variables related to the acceptance of COVID-19 vaccination in parents of children with acute lymphoblastic leukemia (CALLS), and deep into the oncologist-parents relationship that affect that decision. Among several variables, the research found that the most relevant factors influencing the acceptance of COVID vaccination were “with recommendation” and “with a high WAI-SR working alliance.”

 (1) The manuscript is well written, and it is easy to read and understand. Nevertheless, the propensity score-matched (PSM) analysis can be difficult to understand for “average medical doctors,” so I would recommend explaining the basics, and how interpreting the results of PSM simply in the introduction or in the material and methods. If Figures 1 and 2 used the PSM technique, this should be indicated in the legend.

 (2) Could you please provide a more medical background of acute lymphoblastic leukemia (ALL)? What is ALL? How is it diagnosed? Treatment? Prognosis? How does ALL affect the immune system? Is ALL associated with immunosuppression? Etc.

 (3) Could you please provide some basic information about the SARS-CoV-2? Origin, mechanism of action, effects on the body, prognosis, treatment, etc.

 (4) Could you please provide some basic information on how the COVID vaccines work? Which type of covid vaccine was recommended in this research?

 (5) Can the virus infect ALL tumoral cells?

 (6) Regarding “The oncologists of the participants were blind to the research protocol” (line 123). Could you please explain how they were blinded about the protocol?

 (7) Line 124, regarding “does your family agree, ….” Was this question made to the child with leukemia, or to the patients? I thought the question was made to the family.

 (8) Could you please use the word “cross-sectional observational study”

 (9) In Table 1, the income is in Yuan. Could you please also show it in $?

 (10) In Table 1, there are 3 groups: agree, disagree, and hesitant. Although Table 2 seems to be the most important table. Why not also compare the three groups based on each of the factors (variables)? Could you please add the p values? You could calculate a chi-square (3x2 table).

 (11) In the line 212. Next to the “p < 0.001.” Could you please add the corresponding odds ratio (OR) of 3.17.

 (12) In the legend of Figure 1. Could you please add an explanation of what the “recommendation” and “control” group mean?. For example, the text of lines 208-210: “A total of 118 parents (27.8%) received detailed recommendations (recommendation group, RG) from their treating oncologists, and the other 306 parents (72.2%) did not receive the recommendation (control group, CG)”.

 (13) I understand that the results of Figure 1 come after doing the propensity score-matched approach. Is that correct?

 (14) Could you please calculate a univariate binary logistic regression to predict “recommendation” vs “control”, having as predictors the other variables of Table 1?

 (15) Regarding Figure 1. Why the recommendation influences the covid acceptance, but does not change for other vaccines?

 (16) Is the Working Alliance Inventory-Short Revised (WAI-SR) variable only evaluable in the recommended group?

Author Response

Point-by-point response to reviewer:

Thank you very much for the reviewer's meticulous comments and suggestions. We have made modifications one by one, and the replies to the reviewer's comments are uniformly marked with yellow shading in the paper. The details are as follows:

 (1) The manuscript is well written, and it is easy to read and understand. Nevertheless, the propensity score-matched (PSM) analysis can be difficult to understand for “average medical doctors,” so I would recommend explaining the basics, and how interpreting the results of PSM simply in the introduction or in the material and methods. If Figures 1 and 2 used the PSM technique, this should be indicated in the legend.

Answer

Thank you for your advice. As for PSM, we have added relevant explanations in the "statistical analysis" section of the paper. Revision: PSM aims to balance uncontrolled baseline variables in real-world settings that incite bias to statistical test results. Propensity scores were calculated with all baseline variables in regression models. A greedy nearest neighbor matching method was adopted to match participants by such scores. As such, participants were matched to mimic randomized settings in clinical trials and were referred as quasi-randomization. This method may reduce selection bias which is intrinsic to real-world study.. (line 177-182) 

In addition, the legends in Figures 1 and 2 have been modified accordingly: “The comparison of COVID-19 vaccination acceptance and PACV scores between the recommendation group and control group (both in matched samples). (Figure 1) (line 258-259) and “The comparison of parental wiliness against COVID-19 vaccine and non-COVID vaccines between the alliance group and non-alliance group (both in matched samples)(Figure 2)(line 292-293)。

 (2) Could you please provide a more medical background of acute lymphoblastic leukemia (ALL)? What is ALL? How is it diagnosed? Treatment? Prognosis? How does ALL affect the immune system? Is ALL associated with immunosuppression? Etc.

Answer

Thank you for your advice. Due to the limitations of the text structure and space, we put the introduction of related knowledge background for ALL in the supplementary document:“Acute lymphoblastic leukemia (ALL) is a malignant disease resulting from abnormal proliferation of B-line or T-line cells from bone marrow lymphocytes. ALL has been documented as the most common childhood malignancy, accounting for 25% of all childhood cancers [14]. Immunosuppression in patients with leukemia is either due to a disease state involving clonal amplification of undifferentiated and functionally abnormal lymphoid progenitors. They invade the bone marrow, peripheral blood, and extramedullary sites [15] or are immunocompromised due to chemotherapy-induced immunosuppression. Immunocompromised patients are at high risk for viral reactivation or new viral infections [16].

Cell morphology, immunology, cytogenetics and molecular biology can be used to diagnose ALL. The treatment is usually combined with bone marrow transplantation and chemotherapy, and early treatment [17] can obtain a long-term survival prognosis. In the first wave of treatment, patients with hematological malignancies have a poor prognosis, with a mortality rate of 20-40%. The single or simultaneous activation of latent viruses can have serious consequences [19], so the prevention of viral infection is very important. However, severe immunosuppression is a key issue during or after treatment, for which many parents are hesitant to vaccinate their children against COVID-19.” (Please see Acute lymphoblastic leukemia section in Additional Backgrounds in Supplementary Materials)

  • Could you please provide some basic information about the SARS-CoV-2? Origin, mechanism of action, effects on the body, prognosis, treatment, etc.

Answer

Thank you for your advice. Due to the structure and space limitations of the text, we have included the background introduction on COVID-19 in the supplementary document:“Severe Acute respiratory syndrome Coronavirus 2 (SARS-CoV-2) infection and the resulting disease, COVID-19, first emerged in 2019, and WHO declared a pandemic in March 2020[1]. SARS-CoV-2 uses spikes (S) glycoproteins on the viral envelope to attach itself to respiratory cell surfaces that express host cell transmembrane serine protease 2 (TMPRSS2) mediated angiotensin-converting enzyme 2 (ACE2) receptors and S proteins [2, 3]. In addition to the respiratory tract, ACE2 is also expressed on the cells of different tissues, such as alveolar cells of the lung, muscle cells of the heart, and vascular endothelium. After attachment, it can enter cells and replicate in cells to cause disease [2, 3].

Individuals infected with SARS-CoV-2 exhibit a wide range of heterogeneous clinical manifestations, ranging from asymptomatic cases to severe disease that can lead to death [4]. Those at highest risk of severe illness and death include the elderly and those with pre-existing conditions such as cancer [5-7]. The physiology and pathology of COVID-19 involves complex host-viral interactions of different immune cells and inflammatory molecules. Unbalanced immune responses such as low responsiveness (uncontrolled viral replication) and high responsiveness (disproportionate inflammation) can lead to severe COVID-19[4]. Currently, there is a lack of effective treatment for immunocompromised patients, and the treatment is usually complementary therapy [8]. Therefore, the vaccine as a preventive measure can effectively reduce the risk of death in these patients.” (Please see Severe acute respiratory syndrome coronavirus-2 section in Additional Backgrounds in Supplementary Materials)

 (4) Could you please provide some basic information on how the COVID vaccines work? Which type of covid vaccine was recommended in this research?

Answer

Thank you for your advice. Due to text structure and space constraints, we have included the rationale for COVID-19 vaccines in a supplementary document:Similar to other vaccines, the novel coronavirus vaccine infuses the forged novel coronavirus antigen into the human body to make the immune system recognize the antigen and conduct immune attack, forming immune memory [9]. Specifically, if a person is infected, the vaccine will trigger an immune response that can block or kill the virus. Any subsequent COVID-19 antigens will be effectively recognized and attacked by memory immune cells to prevent damage from COVID-19. Researchers around the world have been working to develop a vaccine for COVID-19 since the outbreak began, with more than 198 vaccines currently in preclinical or clinical development. Frantic efforts in vaccine development have led to several vaccine candidates from multiple platforms that have entered the clinical evaluation stage, including inactivated vaccines, live virus vaccines, recombinant protein vaccines, vector vaccines, and DNA or RNA vaccines [11,12].

Currently approved vaccine types in China include 2 doses of inactivated vaccine (Sinovac and Sinopsin) and 1 dose of adenovirus vaccine (Ad5-nCoV). In phase I/II trials, both vaccines showed good immunogenicity and moderate adverse events in healthy people. Because cancer patients are usually immunocompromised, vaccines that carry live viruses are usually prohibited. Therefore, the COVID-19 vaccine recommended by oncologists in this study usually refers to inactivated vaccines. (Please see COVID-19 vaccines section in Additional Backgrounds in Supplementary Materials)

 (5) Can the virus infect ALL tumoral cells?

Answer

Thank you for your comments. There is no concrete proof that this virus may infect tumoral cells. For specific background on COVID-19, see the answer (3).

 (6) Regarding “The oncologists of the participants were blind to the research protocol” (line 123). Could you please explain how they were blinded about the protocol?

Answer

Thank you for your comments. As the researchers themselves conduct the research in a manner of interviewing parents of CALLS, all oncologists that recommend vaccination were not aware of how they or their patients were analyzed in this research. We have also revised the article: The attending oncologists of the participants were blind to the research protocol, as we retrospectively interviewed the parents without seeking confirmation from their doctor.(line 125-127)

 (7) Line 124, regarding “does your family agree, ….” Was this question made to the child with leukemia, or to the patients? I thought the question was made to the family.

Answer

Thank you for your comments. This question was made to the family. As we wrote in the article: During the outpatient visit, parents of the CALLS underwent semi-structured interviews… (line 119-120)

 (8) Could you please use the word “cross-sectional observational study”

Answer

Than you for your advice. We have made changes in the methods section:“From July 5th, 2021, to November 29th, 2021, consecutive parents of CALLS were enrolled in the cross-sectional observational study in the outpatient clinic settings of pediatric hematology department.” (line 108)

 (9) In Table 1, the income is in Yuan. Could you please also show it in $?

Answer

Thank you for your advice. They are illustrated in Table 1 as you suggested:Based on the exchange rate of RMB against US dollar in 2021, 20,000 yuan in the table = 2,824 $,100000yuan=14120$,200000yuan=28240$(line 225-226)

 (10) In Table 1, there are 3 groups: agree, disagree, and hesitant. Although Table 2 seems to be the most important table. Why not also compare the three groups based on each of the factors (variables)? Could you please add the p values? You could calculate a chi-square (3x2 table).

Answer

Thank you for your advice. Since our main research question is vaccine hesitancy, according to the definition of vaccine hesitancy in international researchers, disagree and hesitation belong to the category of vaccine hesitancy, so we did not carry out inter-group analysis of the three dependent variables in this paper. But according to your suggestion, We did comparison between 3 groups: agree, disagree and hesitant group and p values were represented (see table below). However, we do not think these tables are necessary to be reflected in the paper, so we did not add them in the paper.

Factor

All CALLS Families

Agree

N = 91 (21.4%)

Disagree

N = 165 (38.9%)

Hesitant

 N = 168 (39.6%)

p

Child Characteristics

Sex

Female

194 (45.8%)

34 (37.4%)

84 (50.9%)

76 (45.2%)

0.11

Male

230 (54.2%)

57 (62.6%)

81 (49.1%)

92 (54.8%)

Age at Remission

Mean ± SD

5.99 ± 3.40

6.31 ± 3.94

6.05 ± 2.83

5.76 ± 3.60

0.44

Age at Remission (Category)

< 6

220 (51.9%)

50 (54.9%)

75 (45.5%)

95 (56.5%)

≥ 6

204 (48.1%)

41 (45.1%)

90 (54.5%)

73 (43.5%)

Remission Time (Year)

< 1

92 (21.7%)

20 (22.0%)

34 (20.6%)

38 (22.6%)

0.56

1 – 2

78 (18.4%)

15 (16.5%)

32 (19.4%)

31 (18.5%)

2 - 3

108 (25.5%)

30 (33.0%)

42 (25.5%)

36 (21.4%)

≥3

146 (34.4%)

26 (28.6%)

57 (34.5%)

63 (37.5%)

School Preparation

Yes

191 (45.0%)

46 (50.5%)

73 (44.2%)

72 (42.9%)

0.48

Not yet

233 (55.0%)

45 (49.5%)

92 (55.8%)

96 (57.1%)

Relapse History

Ever

165 (38.9%)

30 (33.0%)

68 (41.2%)

67 (39.9%)

0.41

Never

259 (61.1%)

61 (67.0%)

97 (58.8%)

101 (60.1%)

Family Characteristics

Age of Mother at Remission

Mean ± SD

32.50 ± 3.87

32.81 ± 4.04

32.58 ± 3.81

32.26 ± 3.84

0.49

Age of Mother at Remission (Category)

< 33

227 (53.5%)

43 (47.3%)

89 (53.9%)

95 (56.5%)

0.42

≥ 33

197 (46.5%)

48 (52.7%)

76 (46.1%)

73 (43.5%)

Education Level

High school or below

224 (52.8%)

45 (49.5%)

83 (50.3%)

96 (57.1%)

0.05

Junior college

149 (35.1%)

37 (40.7%)

53 (32.1%)

59 (35.1%)

Bachelors or above

51 (12.0%)

9 (9.9%)

29 (17.6%)

13 (7.7%)

Annual Family Income, Yuan

< 20,000

33 (7.8%)

7 (7.7%)

14 (8.5%)

12 (7.1%)

0.53

20,000 to 100,000

168 (39.6%)

39 (42.9%)

64 (38.8%)

65 (38.7%)

100,000 to 200,000

60 (14.2%)

16 (17.6%)

17 (10.3%)

27 (16.1%)

> 200,000

163 (38.4%)

29 (31.9%)

70 (42.4%)

64 (38.1%)

Marital Status

Married

363 (85.6%)

77 (84.6%)

142 (86.1%)

144 (85.7%)

0.95

Divorced

61 (14.4%)

14 (15.4%)

23 (13.9%)

24 (14.3%)

Oncologist Recommendation

Yes

118 (27.8%)

44 (48.4%)

13 (7.9%)

61 (36.3%)

< 0.01

No

306 (72.2%)

47 (51.6%)

152 (92.1%)

107 (63.7%)

Swayed by Non-healthcare Information

Yes

175 (41.3%)

36 (39.6%)

78 (47.3%)

61 (36.3%)

0.12

No

249 (58.7%)

55 (60.4%)

87 (52.7%)

107 (63.7%)

 (11) In the line 212. Next to the “p < 0.001.” Could you please add the corresponding odds ratio (OR) of 3.17.

Answer

Thank you for your advice. We added the odds ratio: “ (OR = 3.17, p < 0.001, see Table 2)”(line 233)

 (12) In the legend of Figure 1. Could you please add an explanation of what the “recommendation” and “control” group mean?. For example, the text of lines 208-210: “A total of 118 parents (27.8%) received detailed recommendations (recommendation group, RG) from their treating oncologists, and the other 306 parents (72.2%) did not receive the recommendation (control group, CG)”.

Answer

Thanks for your advice. We have marked it in a legend: Notes: The “recommendation group” means parents who were recommended by the oncologists, and the “control group” means that parents who did not receive recommendation from the oncologists.(line 261-263)

(13) I understand that the results of Figure 1 come after doing the propensity score-matched approach. Is that correct?

Answer

Thank you for your comments. Yes, all comparison results of the figure 1 come after doing the propensity score-matched approach. We have annotated the figure legend in Figure 1: “The comparison of COVID-19 vaccination acceptance and PACV scores between the recommendation group and control group (both in matched samples).(line 258-259)

 (14) Could you please calculate a univariate binary logistic regression to predict “recommendation” vs “control”, having as predictors the other variables of Table 1?

Answer

Thanks for your advice, we did univariate binary logistic regression, the table is as follows. Doctors’ advice only represented personal views and did not involve appraisal of relations with parents or patients, and thus we did not put the table in the manuscript.

Univariate binary logistic regression to predict oncologist recommendation vs control

Variables

P

OR

95% CI of OR

Lower limit

Upper limit

Marital Status

0.92

0.97

0.48

1.95

Relapse History

0.25

0.74

0.44

1.24

Education Level

0.81

0.96

0.67

1.37

Recommended by doctors

0

3.12

1.87

5.2

Annual Family Income

0.68

0.95

0.75

1.21

School Preparation

0.97

1.01

0.61

1.68

Age at Remission

0.63

1.02

0.95

1.1

Sex(male)

0.15

1.46

0.87

2.43

Remission Time (Year)

0.99

1

0.8

1.24

Mother’s Gestating Age

0.97

1

0.81

1.25

 (15) Regarding Figure 1. Why the recommendation influences the covid acceptance, but does not change for other vaccines?

Answer

Thank you for your comments. Because the doctors' advice we collected was only for COVID-19, people were willing to take the vaccine when doctors suggested it. The insignificance of PACV score is to eliminate the possible bias of parents on the vaccine itself, which has nothing to do with doctors' suggestions. We did not collect doctors' suggestions on other vaccines, which may be worth studying in the future.

  • Is the Working Alliance Inventory-Short Revised (WAI-SR) variable only evaluable in the recommended group?

Answer

Thank you for your comments. Yes, because we primarily aim to find whether alliance may affect COVID-19 vaccine recommendation potency, we only evaluated the WAI-SR in the recommended group. We add an explanation in the method section of the paper:The WAI-SR was only collected from parents who was recommended to vaccinate COVID-19 by the attending oncologist.”(line 141-143)
